# A Novel Recombinant Fcγ Receptor-Targeted Survivin Combines with Chemotherapy for Efficient Cancer Treatment

**DOI:** 10.3390/biomedicines9070806

**Published:** 2021-07-12

**Authors:** Chiao-Chieh Wu, Chen-Yi Chiang, Shih-Jen Liu, Hsin-Wei Chen

**Affiliations:** 1National Institute of Infectious Diseases and Vaccinology, National Health Research Institutes, Miaoli 35053, Taiwan; chiaochieh@nhri.edu.tw (C.-C.W.); cycjen@nhri.edu.tw (C.-Y.C.); levent@nhri.edu.tw (S.-J.L.); 2Graduate Institute of Biomedical Sciences, China Medical University, Taichung 406040, Taiwan; 3Graduate Institute of Medicine, Kaohsiung Medical University, Kaohsiung 307378, Taiwan

**Keywords:** cancer vaccine, Fcγ receptor, formyl peptide receptor-like 1 inhibitor, survivin, immunotherapy

## Abstract

Formyl peptide receptor-like 1 inhibitor (FLIPr), an Fcγ receptor (FcγR) antagonist, can be used as a carrier to guide antigen-FLIPr fusion protein to FcγR then enhances antigen-specific immune responses. Survivin, a tumor-associated antigen, is over-expressed in various types of human cancer. In this study, we demonstrate that recombinant survivin-FLIPr fusion protein (rSur-FLIPr) binds to FcγRs, and efficient uptake by dendritic cells in vivo. In addition, rSur-FLIPr alone stimulates survivin-specific immune responses, which effectively suppresses the tumor growth. The antitumor immunities are through TAP-mediated and CD8-dependent pathways. Furthermore, preexisting anti-FLIPr antibody does not abolish antitumor responses induced by rSur-FLIPr immunization. These results suggest that FLIPr is an effective antigen delivery vector and can be repeatedly used. Combination of chemotherapy with rSur-FLIPr treatment reveals a great benefit to tumor-bearing mice. Altogether, these findings suggest that rSur-FLIPr is a potential candidate for efficient cancer therapy.

## 1. Introduction

The goal of a therapeutic cancer vaccine is to target tumor-associated antigens (TAAs) to then elicit the patient’s own immune system against the cancer cells. However, stimulating robust immune responses against self-TAAs is still difficult to achieve. Dendritic cells (DCs) are the most potent professional antigen-presenting cells and play a key role during the initiation of antigen-specific immune responses. They patrol the body to capture antigens and then process and present them to monitor pathogen infection and cell transformation [1,2]. Therefore, targeting antigen to DCs is an effective way to elicit antigen-specific immune responses.

An antibody binds to an antigen to form an immune complex (IC). Fcγ receptors (FcγRs) expressed on the surface of DCs play an important role in internalizing IC and modulating immune responses [3]. Antigens captured by antigen-presenting cells via FcγRs have been shown to efficiently trigger antigen-specific immune responses [4,5,6,7]. However, an antibody recognizing a specific antigen and forming ICs for facilitating phagocytosis as a vaccine strategy is not easy [8]. Formyl peptide receptor-like 1 inhibitory protein (FLIPr), secreted by *Staphylococcus aureus*, is a potent FcγR antagonist that can bind to FcγRs [9,10]. Using this property of FLIPr, an antigen-FLIPr fusion protein can be similar to an IC, which contains antigen and an FcγR-binding ability. We have demonstrated that the immunization of mice with an antigen-FLIPr fusion protein facilitates antigen processing and presentation in both MHC class I and class II pathways, which then further enhances antigen-specific immune responses [11]. These novel findings indicate that FLIPr is a potent vector for antigen delivery and can be applied to cancer vaccine development.

Survivin belongs to the inhibitor of apoptosis protein family and plays a key role in the regulation of mitosis and apoptosis [12,13]. The expression of survivin in cells is usually detected in the embryonic lung and fetal organs at the developing stages but is rarely detected in terminally differentiated adult tissues except for the thymus, placenta, CD34+ stem cells, and basal colon epithelial cells [13,14,15]. However, survivin is broadly expressed in most human cancer cells, including lung, breast, pancreatic, and colon carcinomas; prostate, ovarian, and soft tissue sarcomas; brain tumors; melanoma; neuroblastoma; and hematologic malignancies [16,17,18,19,20,21,22,23,24,25,26]. In addition, during blood vessel formation, survivin is highly expressed in actively dividing endothelial cells [27,28,29], which then promotes vascular endothelial growth factor-induced tumor angiogenesis for nutrient transportation [30]. Therefore, the overexpression of survivin provides great benefits for tumor formation. According to these properties, survivin may be a universal tumor-associated antigen for cancer immunotherapy.

In this study, survivin fused with FLIPr was expressed in an *Escherichia coli*-based system. We hypothesized that the recombinant survivin-FLIPr fusion protein (rSur-FLIPr) is a broad-spectrum cancer vaccine candidate with the ability to target FcγRs, which is able to elicit survivin-specific immune responses against survivin-expressing cancers. Our results indicated that rSur-FLIPr alone or combined with chemotherapy is feasible for cancer immunotherapy.

## 2. Materials and Methods

### 2.1. Mice

Female C57BL/6 mice, 5~12 weeks of age, were purchased from the National Laboratory Animal Center, Taipei, Taiwan. TAP1-deficient mice were purchased from Jackson Lab. All of the mice were housed and bred at the Laboratory Animal Center of the National Health Research Institutes (NHRI) in Taiwan. All animal studies were approved and were performed in compliance with the guidelines of the Animal Committee of the NHRI.

### 2.2. Construction of Expression Vectors

Restriction enzymes and the ligase for plasmid construction were purchased from New England Biolabs, Inc. (Beverly, MA, USA). Based on the amino acid sequence of human survivin (accession number AAC51660) and FLIPr (accession number BAB57318), the DNA sequence encoding rSur-FLIPr was optimized for *E. coli* codon usage and fully synthesized by the Genomics Co. (New Taipei City, Taiwan). The forward primer (5′-GGAATTCCATATGATGGGCGCGCCGACCCTGCCGCC-3′, NdeI site is underlined) combined with reverse primer (5′-CACGAGCTCGAGATCCCAATAAATGCTATC-3′, XhoI site is underlined) were used to amplify the synthetic DNA of rSur-FLIPr. The PCR product was then cloned into the NdeI and XhoI sites of the expression vector pET-22b(+) (Novagen, Madison, WI, USA) to produce the plasmid pSur-FLIPr. As a result, the C-terminus of rSur-FLIPr contained a hexahistidine tag (His-tag). The construction of rSur expression vectors and their purification were described previously [31].

### 2.3. Production and Purification of rSur and rSur-FLIPr

All experimental chemicals were acquired from Sigma-Aldrich (St. Louis, MO, USA) and Merck (Darmstadt, Germany). To express the protein, *E. coli* BL21 (DE3) (Invitrogen, Carlsbad, CA) was transformed with pSur-FLIPr. The transformed cells were cultured at 25 °C overnight. The overnight culture (40 mL) was amplified to 1 L in a 2 L shake flask and incubated at 37 °C for 2.5 h before induction. When the culture OD_600_ reached 0.3, IPTG (1 mM) was added to induce protein expression by incubation at 37 °C for 4 h. To purify rSur-FLIPr, the harvested cells were disrupted in a French press (Constant Systems, Daventry, UK) at 25 Kpsi in homogenization buffer [20 mM Tris (pH 8.0), 50 mM sucrose, 500 mM NaCl, and 10% glycerol]. The cell lysate was clarified by centrifugation (119,000× *g* for 40 min). Most of the rSur-FLIPr was present in inclusion bodies. The rSur-FLIPr was then solubilized with extraction buffer [20 mM Tris (pH 8.0), 50 mM sucrose, 500 mM NaCl, 10% glycerol, and 3 M guanidine hydrochloride]. The extraction was clarified by centrifugation (119,000× *g* for 40 min). The supernatant of the extracted fraction was loaded onto immobilized metal affinity chromatography (IMAC) columns (BIO-RAD, Hercules, CA, USA, 2.5 cm i.d. × 10.0 cm) containing 20 mL Ni-NTA resin (Qiagen, San Diego, CA, USA) to purify rSur-FLIPr. The column was washed with the extraction buffer, the extraction buffer containing 20 mM imidazole, and then with a 200-fold column volume of 10 mM Na_2_HPO_4_ containing 0.1% Triton X-114 (pH 9.6) to remove the endotoxin. Next, the column was washed with 10 mM Na_2_HPO_4_ without 0.1% Triton X-114 to remove the residual detergent, and rSur-FLIPr was eluted with 10 mM Na_2_HPO_4_ containing 300 mM imidazole (pH 9.6). The eluted rSur-FLIPr was dialyzed against 10 mM Na_2_HPO_4_ 3 times and for at least 6 h each time. The residual endotoxin in the purified rSur-FLIPr was evaluated by the Limulus amebocyte lysate (LAL) assay (Associates of Cape Cod, Inc., Cape Cod, MA, USA), which was found to be less than 30 EU/mg. After dialysis, the rSur-FLIPr was lyophilized and stored at −20 °C. The purified proteins were analyzed by tricine-PAGE and immunoblotted with anti-survivin (R&D Systems, Minneapolis, MN, USA) or anti-FLIPr (serum from the C57BL/6 mice immunized with recombinant FLIPr) antibodies.

### 2.4. Capture Enzyme-Linked Immunosorbent Assays

The procedures for the biotinylation of rSur or rSur-FLIPr were described by the EZ-Link NHS-PEG4-Biotinylation Kit (Thermo Fisher Scientific, Rockford, IL, USA). Fcγ receptors were purchased from Sino Biological, Inc. (Chesterbrook, PA, USA) or ACROBiosystems (Newark, DE, USA). Mouse Fcγ receptor-1, -2b, -3, or -4; or human Fcγ receptor-1, -2b, -2a (H167), -2a (R167), -3a (V176), -3a (F176), -3b (NA1), or -3b (NA2) was coated on 96-well plates (0.2 μg/well). Nonspecific binding was blocked by 5% skim milk in PBS. A serial dilution (3-fold, starting at 0.5 mM) of biotin-conjugated rSur or rSur-FLIPr was added to each well and then incubated at room temperature for 2 h. After washing, HRP-conjugated streptavidin was added to detect the binding proteins. A substrate, 3, 3′, 5, 5′-tetramethylbenzidine (TMB), was added and incubated for 10 min for color development. The absorbance was measured with an ELISA reader at 450 nm.

### 2.5. Uptake of rSur-FLIPr in In Vivo Assays

The procedures of labeling rSur or rSur-FLIPr were described in the manual of the Alexa Fluor^TM^ 647 Protein Labeling Kit (Invitrogen; Thermo Fisher Scientific, Waltham, MA, USA). The groups of mice were injected with Alexa 647-labeled rSur or rSur-FLIPr in the hind foot pads (100 μg/foot pad) or with PBS as a control. The cells of the inguinal lymph nodes were harvested at 24 and 40 h after injection. According to the protocol of the LIVE/DEAD^®^ Fixable Dead Cell Stains (Thermo Fisher Scientific, Waltham, MA, USA), the cells were stained with LIVE/DEAD^®^ Fixable Dead Cell Stains and with CD19 (6D5), CD3e (145-2C11), NK1.1 (PK136), Ly6G (1A8), CD11c (N418), and MHCII (M5/114.15.2) antibodies. The data acquisitions were analyzed using the Attune NxT Flow Cytometer (Invitrogen).

### 2.6. Enzyme-Linked Immunospot (ELISPOT) Assays

The splenocytes were prepared one week after the last immunization. IFN-γ-producing cells in the spleen were determined using a mouse IFN-γ ELISPOT kit (PB Pharmingen, San Diego, CA, USA) according to the manufacturer’s instruction. As described previously [11], the capture antibodies were coated on 96-well plates with PVDF membranes (Millipore, Burlington, MA, USA) and then incubated at 4 °C overnight. After washing with PBS, the plates were blocked with RPMI medium supplemented with fetal bovine serum (10%) for 1 h to prevent nonspecific binding in the later steps. The groups of the splenocytes (5 × 10^5^ cells/well) were seeded into the plates with Sur_21-29_ (TFKNWPFLE) and Sur_57-64_ (CFFCFKEL) peptides in triplicate wells. In parallel, RAH control peptide (RAHYNIVTF, derived from E7 of HPV) and media (no stimulation) were included as controls. The splenocytes were discarded from the plates by washing three times with 0.05% (*w*/*v*) Tween 20 in PBS after incubation at 37 °C in a 5% CO_2_ humidified incubator for 3 days. The biotinylated detection antibody was added to the wells (0.1 mL/well), after which the plates were incubated at room temperature for 2 h. After repeating the washing steps outlined above and adding the avidin-horseradish peroxidase complex reagent, the plates were incubated at room temperature for 45 min. The plates were washed three times with 0.05% (*w*/*v*) Tween 20 in PBS and then three times with PBS alone. Staining solution (3-amine-9-ethylcarbazole, Sigma-Aldrich, St. Louis, MO, USA) was added to the wells (0.1 mL/well) to develop the spots. After 1 h, the plates were placed under tap water to stop the reaction. The spots were determined by an ELISPOT reader (Cellular Technology Ltd., Shaker Heights, OH, USA).

### 2.7. Tumor Models

EG7 cells (Bioresource Collection and Research Center, Hsinchu, Taiwan, BCRC Number: 60418) were cultured in RPMI 1640 medium supplemented with 10% (*v*/*v*) heat-inactivated fetal bovine serum, L-glutamine (2 mM), sodium pyruvate (1 mM), HEPES (10 mM), G418 (0.4 mg/mL), 2-mercaptoethanol (0.05 mM), and penicillin/streptomycin (50 units/mL) at 37 °C under 5% CO_2_. B16F10 cells (Bioresource Collection and Research Center, Hsinchu, Taiwan, BCRC Number: 60031) were cultured in DMEM medium supplemented with 10% (*v*/*v*) heat-inactivated fetal bovine serum and penicillin/streptomycin (50 units/mL) at 37 °C under 5% CO_2_. These tumor cells were harvested and washed with PBS. The mice were subcutaneously (s.c.) inoculated with 5 × 10^4^ EG7 cells or 1 × 10^5^ B16F10 cells in 100 μL of PBS in the left flank on day 0. For the prophylactic model, the groups of wild type or TAP-deficient mice were immunized twice on day -21 and -7 with antigens. For the therapeutic model, the groups of mice were immunized twice on day 3 and 10 with the antigens. To deplete the subpopulation of CD4^+^ or CD8^+^ cells in mice, the mice were treated intraperitoneal (i.p.) by injection with 0.5 mg of the rat anti-mouse CD4 antibody (clone RM4–5, Biolegend) or CD8 antibody (clone 53–6.72, Biolegend) to deplete the CD4^+^ or CD8^+^ cells before day one of the EG7 challenge. A total of 0.5 mg of rat IgG2a (Biolegend) were used as a control antibody in the experiments. Tumor growth was monitored by visual inspection and palpation. The tumor growth and survival rate were monitored. The tumor size was measured with a caliper, and the tumor volume was estimated by the formula V = length × width × width/2. The mice were sacrificed when the tumor volume reached 3000 mm^3^.

### 2.8. Cyclophosphamide (CTX) Combined Therapy in Tumor Model

Female C57BL/6 mice were subcutaneously implanted with EG7 cells (5 × 10^4^ cells per mouse) or B16F10 cells (1 × 10^5^ cells per mouse) into the left flank prior to chemotherapy or immunization with CTX (Sigma-Aldrich, St. Louis, MO, USA) or rSur-FLIPr. The tumor-bearing mice were administered CTX (3 mg in 100 µL PBS/per mouse) via i.p. injection on day 10, 12, and 14 and then s.c. administered two doses of rSur-FLIPr (30 µg per mouse) on day 17 and 24. For the rechallenge model, the survivors that had no detectable EG7 tumor growth 100 days after treatment with CTX and rSur-FLIPr were used, with naïve mice used as a control; the mice were inoculated with 5 × 10^4^ EG7 cells in the right flank.

### 2.9. Measurement of the Antibody Titers

C57BL/6 mice were preimmunized by subcutaneous injection without/with 30 µg rFLIPr plus 50 µL aluminum phosphate (Brenntag Biosector, Frederikssund, Denmark) on day -32 and -18. The blood samples were collected by submandibular blood collection on day -32 and -4, after which the blood was centrifuged, and the serum collected. The levels of rFLIPr IgG in the serum samples were determined by titrating the samples. The sera were diluted by 3-fold serial dilution at a 30-fold dilution of the serum samples. Briefly, purified rFLIPr was coated onto 96-well ELISA plates overnight. Bound IgG was detected with HRP-conjugated goat anti-mouse IgG. After the addition of TMB, the absorbance was measured with an ELISA reader at 450 nm. The ELISA end-point titers were defined as the serum dilution that produced an OD value of 0.2. The serum dilution was obtained from the titration curve by interpolation.

### 2.10. Data Analysis

The Kruskal–Wallis test with Dunn’s multiple comparison was used to compare the differences for more than two groups. The statistical significance of the tumor study was determined using the log-rank (MauteeCox) test. The difference between two groups was determined using the Mann–Whitney test. Statistical analysis was performed using GraphPad Prism software version 5.02 (GraphPad Software, San Diego, CA, USA). Differences with a *p* < 0.05 were considered to be statistically significant.

## 3. Results

### 3.1. Production and Characterization of rSur-FLIPr

To study the capability of FLIPr-delivered survivin for cancer immunotherapy, recombinant survivin (rSur) and rSur-FLIPr were prepared from an *E. coli*-based system. Both purified proteins were analyzed by 10% tricine-PAGE, followed by staining with Coomassie Blue or further examination by immunoblotting with anti-survivin or anti-FLIPr antibodies (Figure 1a). Both rSur and rSur-FLIPr were detected by anti-survivin antibodies. However, rSur-FLIPr, but not rSur, was recognized by anti-FLIPr antibodies. These results indicate that the purified recombinant proteins are rSur and rSur-FLIPr, respectively.

Next, we examined the interaction between rSur-FLIPr and FcγRs by a capture ELISA. The rSur-FLIPr was captured by various mouse (Figure 1b) and human (Figure 1c) FcγR isoforms in a dose-dependent manner. In contrast, there was no or slight interaction between rSur and FcγRs, even when adding rSur at more than 10 nM. These results indicate that rSur-FLIPr has the ability to bind to different mouse and human FcγR isoforms.

### 3.2. Targeting of rSur-FLIPr to Dendritic Cells and Increasing the Efficiency of Cross-Presentation

Professional APCs, such as DCs, patrol the peripheral tissues and then move to lymph nodes after capturing antigens in order to trigger adaptive immune responses. To evaluate whether rSur-FLIPr was delivered to DCs in vivo, the lymphocytes were harvested from the draining lymph nodes 24 or 40 h after injection of the mice with 100 µg of Alexa 647-labeled rSur-FLIPr, rSur, or PBS. The frequencies of the antigen harbored in the DCs were analyzed by flow cytometry. The gating strategy for the DC population and the representative results are shown in Figure 2a. Although the DCs quickly captured rSur or rSur-FLIPr at 24 h after injection, the frequencies of the antigen harbored in the DCs in the draining lymph nodes were further elevated in the rSur-FLIPr-injected mice but not in the rSur-injected mice (Figure 2b). These results suggest that rSur-FLIPr is superior to rSur and was efficiently directed to DCs in vivo.

To further investigate whether the immune responses elicited by rSur-FLIPr in vivo were superior to those elicited by rSur, the mice were immunized twice at a 2-week interval with rSur-FLIPr, rSur, or PBS. One week after the last immunization, the splenocytes from the immunized mice were harvested, after which survivin-specific CD8^+^ T cell responses were evaluated by ELISPOT. Two CD8-restricted epitopes were used for the stimulation. One epitope was a peptide derived from survivin_21-29_. We previously discovered that survivin_17-30_ contained a CD8-restricted epitope [31] and further refined it to survivin_21-29_ (Appendix A). The other epitope was a peptide derived from survivin_57-64_ [32]. Both survivin_21-29_ and survivin_57-64_ are conserved between mouse and human survivin. The frequencies of the IFN-γ-secreting cells were significantly elevated upon stimulation with survivin_21-29_ or survivin_57-64_ in the mice immunized with rSur-FLIPr but not in the mice immunized with rSur or PBS. However, the frequencies of the IFN-γ-secreting cells were not significantly increased after stimulation with the control peptide or without stimulation in all three groups of mice (Figure 3). These results indicate that immunization with rSur-FLIPr alone can induce CD8^+^ T cell responses in mice.

### 3.3. Induction of Antitumor Immunity by Immunization with rSur-FLIPr Is Mediated by CD8^+^ T Cells and through a TAP-Dependent Pathway

In view of the excellent immune response induced by vaccination with rSur-FLIPr alone, we next investigated the in vivo antitumor effect of rSur-FLIPr on vaccinated mice. The experimental schemes for immunization and tumor challenge are shown. One week after the last immunization, EG7 cells were injected into the immunized mice. Tumor growth was inhibited in the mice that received the rSur-FLIPr vaccination compared with mice that received the PBS vaccination but not in mice that received the rSur vaccination (Figure 4a).

To determine whether CD4^+^ or CD8^+^ cells contributed to the antitumor immunity, anti-CD4 or anti-CD8 antibodies were intraperitoneally injected into the rSur-FLIPr-immunized mice one day before tumor inoculation. The inhibition of tumor growth in the mice that received the rSur-FLIPr vaccination was abolished when the mice were depleted of CD8^+^ cells with the anti-CD8 antibodies (Figure 4b). In contrast, the rSur-FLIPr-immunized mice injected with the anti-CD4 or isotype control antibodies still maintained the capacity of tumor growth inhibition. These results indicate that the CD8^+^ cells are the primary population to mediate the in vivo antitumor responses in the rSur-FLIPr-immunized mice.

We further investigated whether the TAP-dependent MHC class I pathway was involved in the induction of antitumor effects derived from the rSur-FLIPr immunization. Wild type and TAP-deficient mice were immunized with rSur-FLIPr in parallel. Tumor growth in the rSur-FLIPr-immunized TAP-deficient mice was comparable to the PBS-immunized TAP-deficient mice. In contrast, the rSur-FLIPr-immunized wild type mice still maintained the capacity of tumor growth inhibition (Figure 4c). These results indicate that the antitumor effect induced by rSur-FLIPr is via the TAP-dependent pathway.

### 3.4. The Combination of Chemotherapy with rSur-FLIPr Increases the Therapeutic Potential and Prolongs Animal Survival

To assess the therapeutic potential by treatment with rSur-FLIPr, tumor-bearing mice were injected with PBS, rSur, or rSur-FLIPr on days 3 and 10 after tumor inoculation. Tumor growth was suppressed in the mice treated with rSur-FLIPr in comparison to the mice treated with PBS or rSur in both the EG7 (Figure 5a) and B16F10 (Figure 5c) models. In the EG7 models, the median survival times were 21 and 23 days for the PBS- and rSur-treated mice, respectively. Significantly, the median survival times were prolonged to 37 days when the mice were treated with rSur-FLIPr (Figure 5b). In mice injected with B16F10, the median survival times were 21, 16, and 28 days when the mice were treated with PBS, rSur, and rSur-FLIPr, respectively (Figure 5d). These results suggest that rSur-FLIPr has therapeutic potential in both EG7 and B16F10 models.

Chemotherapy is an important modality to treat malignancies. We further evaluated the antitumor capacity by combining chemotherapy (cyclophosphamide, CTX) with rSur-FLIPr. The tumor-bearing mice were treated with CTX at day 10, 12, and 14 or treated with rSur-FLIPr at day 17 and 24 after tumor inoculation. The mice treated with PBS served as the controls. After treatment with CTX alone, the EG7 (Figure 5e) and B16F10 (Figure 5g) tumor volumes had shrunk, and the survival times of the mice were prolonged (Figure 5f,h). CTX plus rSur-FLIPr treatment further inhibited tumor growth and extended the survival time. Notably, there was no detectable EG7 tumor growth in the CTX plus rSur-FLIPr-treated mice 100 days after tumor inoculation. These mice were inoculated with the EG7 tumor cells again. The tumor growth of EG7 was still restrained (Figure 5i). Five out of nine mice survived tumor free for more than 2 months (Figure 5j).

### 3.5. Preexisting Anti-FLIPr Antibody Does Not Diminish the Antitumor Capacity Induced by rSur-FLIPr

To assess whether the preexisting anti-FLIPr antibodies interfered with the antitumor capacity induced by rSur-FLIPr treatment, groups of mice were immunized twice at a two-week interval with rFLIPr plus AlPO_4_ or PBS prior to EG7 inoculation. Both groups of mice were randomly divided into two subgroups and then treated with rSur-FLIPr or PBS, respectively. The experimental scheme is shown in Figure 6a. The anti-FLIPr antibodies were successfully elicited after immunization with rFLIPr plus AlPO_4_ (Figure 6c) but not after PBS immunization (Figure 6b). Before inoculation of the tumor cells, the anti-FLIPr antibody titers were found to be equivalent between the subgroups of the rSur-FLIPr- and PBS-treated mice (Figure 6c). The antitumor capacities in the rSur-FLIPr-treated mice with preexisting anti-FLIPr antibodies (Figure 6e) were similar to those in mice without preexisting anti-FLIPr antibodies (Figure 6d). These results suggest that preexisting anti-FLIPr antibodies do not block the antitumor capacity induced by rSur-FLIPr immunization.

## 4. Discussion

CD8^+^ T cells are the immune cells of choice for targeting cancer [33]. Induction of the cytotoxic T cell response is one of the rational immunotherapy strategies and is a key index for evaluating the efficacy of cancer vaccines. In this study, we showed that rSur-FLIPr alone is able to stimulate survivin-specific CD8^+^ T cell responses (Figure 3) and induce antitumor ability (Figure 4a). The capability of tumor growth inhibition in mice immunized with rSur-FLIPr is eliminated when CD8^+^ T cells are depleted prior to tumor inoculation (Figure 4b). These results indicate that survivin-specific CD8^+^ T cells play an important role in inhibiting tumor growth.

Professional APCs, such as DCs, patrol the peripheral tissues and then move to the lymph nodes after capturing antigens in order to trigger adaptive immune responses. Antigen-antibody ICs can be captured by DCs through FcγRs and then regulate immune responses [3,34]. ICs are more efficient than antigen alone in the facilitation of DCs to stimulate antigen-specific T cell responses [3,34]. DCs are the key cells to mediate cross-presentation of exogenous antigens [35,36]. In addition, ICs can enter the cross-presentation pathway to enhance CD8^+^ T cell responses via the TAP-dependent pathway [37,38,39]. TAP plays a critical role in transporting short peptides into the endoplasmic reticulum for their subsequent assembly with MHC I molecules for the activation of CD8^+^ T cells [40]. We demonstrate that rSur-FLIPr but not rSur binds to various mouse and human FcγR isoforms in vitro (Figure 1b,c). Furthermore, rSur-FLIPr is superior to rSur in targeting to DCs and is efficiently captured by DCs in vivo (Figure 2). These results support that FLIPr fused with survivin can guide rSur-FLIPr to FcγRs and increase survivin uptake by DCs. Survivin captured by DCs in the rSur-FLIPr form may be routed to the cross-presentation pathway, which is similar to the form of the survivin-antibody IC. Without TAP, rSur-FLIPr is unable to elicit robust antitumor responses (Figure 4c). Altogether, rSur-FLIPr behaves similar to survivin-antibody ICs to enhance survivin-specific immune responses.

Since rSur-FLIPr binds to various FcγR isoforms (Figure 1b,c), it is very likely that rSur-FLIPr can be directed to all other FcγR expression cells. However, only professional APCs are able to activate naïve T cells. In our previous studies [11], DCs (CD11c^+^ subset), but not the CD11c^−^ subset, in the lymph nodes of injected sites obtained from mice immunization with recombinant OVA-FLIPr fusion protein activated naïve OT-1 and OT-2 T cells. These results suggest that DCs, but not other FcγR expression cells, are the primary cells to trigger antigen-specific immune responses.

Therapeutic vaccines are used for strengthening the patient’s own immune responses to treat the late-stage diseases of cancer [41], but tumor-associated antigens are self-antigens with a low immunogenicity [42], and this is a critical issue that must be addressed. In the therapeutic models, when we immunized the tumor-bearing mice with rSur-FLIPr, the tumor growth was suppressed (Figure 5a,c) and animal survival was prolonged (Figure 5b,d) in the EG7 and B16F10 models. No significant therapeutic benefits were found in the mice treated with rSur. The results presented here delineate the feasibility of cancer immunotherapy using FLIPr as an antigen delivery system. Based on the present outcome, rSur-FLIPr alone may not be sufficient to effectively treat cancer. Immunotherapy, together with surgery, radiotherapy, and chemotherapy, have been considered the four pillars of cancer treatment. To extend the impact of rSur-FLIPr to more patients and a broader range of cancers, a combination with other approaches will be critical. Cyclophosphamide is one of the most successful and widely utilized antineoplastic drugs [43]. A combination of CTX with rSur-FLIPr increased the therapeutic effects, not only suppressing tumor growth but also prolonging survival time in tumor-bearing mice (Figure 5e–h). Significantly, rSur-FLIPr immunization after the CTX treatments completely inhibited the tumor growth in the EG7 model (Figure 5e,f). The tumor-free mice still maintained the capabilities to suppress tumor recurrence even 100 days after the first tumor inoculation (Figure 5i,j). These results suggest that long-lasting antitumor immunities have been elicited in the survivors. Collectively, our studies show that rSur-FLIPr can be easily combined with regular chemotherapy to provide a great benefit to the host. It is worth moving forward into further clinical studies.

FLIPr is produced by *Staphylococcus aureus*, which is both a commensal bacterium and a human pathogen, and approximately 30% of the human population is colonized with *S. aureus* [44,45]. It is likely that many cancer patients have existing anti-FLIPr antibodies. Preexisting immunity against vaccine vectors in the host may have harmful effects on the subsequent immune response to a vectored antigen [46,47]. To address the effect of antitumor responses via rSur-FLIPr vaccination on the preexisting anti-FLIPr antibodies, we immunized mice with rFLIPr plus AlPO_4_ or PBS before tumor inoculation. As expected, anti-FLIPr antibodies were induced in rFLIPr plus AlPO_4_-immunized mice (Figure 6c). In rSur-FLIPr-treated mice, the antitumor response was not affected in mice with preexisting anti-FLIPr antibodies. These results support the hypothesis that preexisting anti-FLIPr antibody is not harmful for the subsequent FLIPr-based immunization (Figure 6d,e). In addition, our results also indicate that antigen-FLIPr can be repeatedly used with the same or different antigens without abolishing the antigen-specific immune responses. This is dissimilar from viral vectors, where preexisting anti-vector immunity may diminish vector efficacy [48].

## 5. Conclusions

TAAs are poorly immunogenic in nature, and an immunostimulatory adjuvant is essential for the generation of an effective immune response [41]. The rSur-FLIPr alone could induce antitumor immunities, which illustrated that the recombinant immunogen-FLIPr technology can easily be applied to other antigens without additional modification or adjuvant. This provides a new platform for the development of successful immunotherapies using protein-based candidates and will hopefully bolster efforts to yield safe and effective vaccines for human use.

## Figures and Tables

**Figure 1 biomedicines-09-00806-f001:**
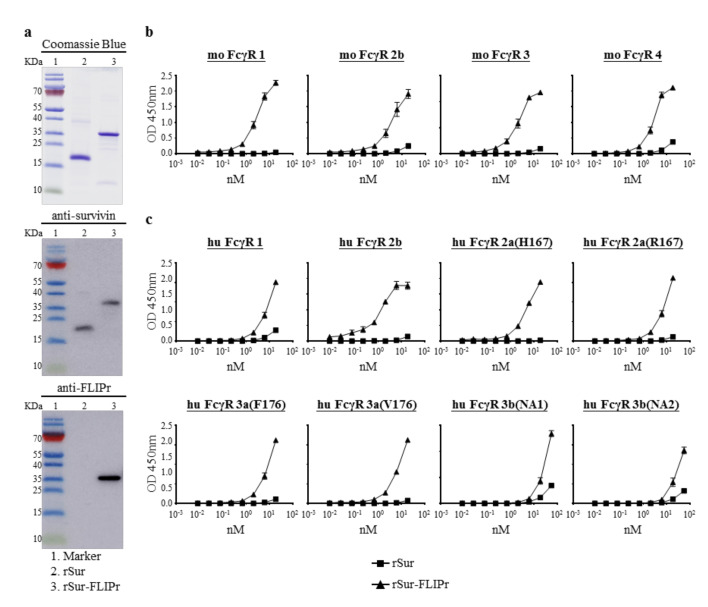
Production and characterization of rSur and rSur-FLIPr. (**a**) The purified proteins were examined by 10% tricine-PAGE with Coomassie Blue staining or examined by immunoblotting with anti-survivin and anti-FLIPr antibodies. Lane 1, marker; lane 2, rSur; lane 3, rSur-FLIPr. Binding ability of rSur-FLIPr to (**b**) mouse and (**c**) human Fcγ receptors. Various Fcγ receptor isoforms were coated on 96-well plates (0.5 μg/well). A serial dilution of biotin-conjugated rSur or rSur-FLIPr was added to each well and incubated at room temperature for 2 h. The binding proteins were detected by adding HRP-conjugated streptavidin. A substrate, TMB, was added for color development. The absorbance was measured with an ELISA reader at 450 nm. The data represent the means ± SE of the mean from two independent experiments.

**Figure 2 biomedicines-09-00806-f002:**
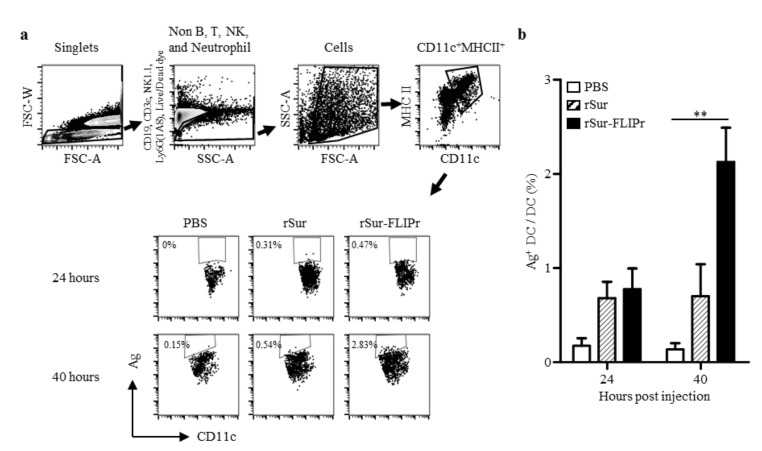
rSur-FLIPr is efficiently captured by dendritic cells. Groups of C57BL/6 mice were injected with Alexa 647-labeled rSur or rSur-FLIPr in the hind foot pads (100 μg/foot pad). Mice injected with PBS were used as controls. Draining lymph nodes were harvested at 24 and 40 h after injection. (**a**) Gating strategy of the DC population in mouse lymph nodes. Single cells were gated by FSC-W/SSC-A. Dead cells were removed from the analysis using LIVE/DEAD^®^ fixable dead cell stains. B cells, T cells, NK cells, and neutrophils were excluded from the analysis by staining with CD19, CD3e, NK1.1, and Ly6G (1A8) antibodies. CD11c^+^MHCII^+^ cells were further analyzed for CD11c and labeled antigen. (**b**) MHCII^+^CD11c^+^ cells were analyzed for the expression of labeled antigens. The results were pooled from three independent experiments and expressed as the mean ± SE of the mean (*n* = 8). The statistical significance was determined using the Kruskal–Wallis test with Dunn’s multiple comparison test. ** *p* < 0.01.

**Figure 3 biomedicines-09-00806-f003:**
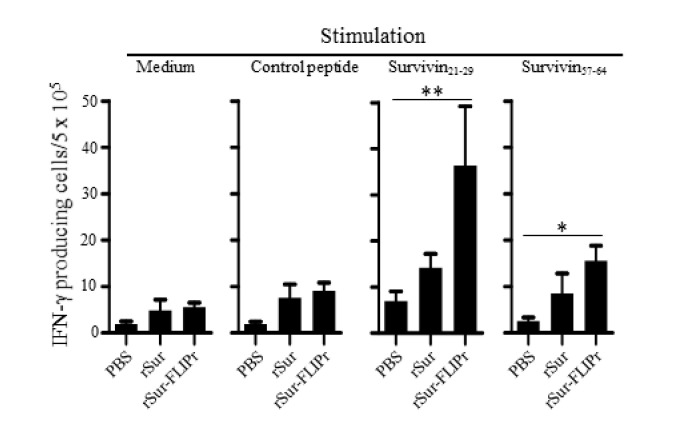
Immunization with rSur-FLIPr induces Sur-specific CD8^+^ T cell responses. Groups of C57BL/6 mice were immunized with rSur or rSur-FLIPr twice at a two-week interval (30 µg per dose). Mice injected with PBS served as the controls. Seven days after the second immunization, splenocytes were stimulated with survivn_21-29_, survivn_57-64_, or control peptides for 72 h in anti-INF-γ-coated 96-well ELISPOT plates. The IFN-γ-secreting spots were examined using an ELISPOT reader. The results were pooled from two independent experiments and expressed as the mean ± SE of the mean (*n* = 6). The statistical significance was determined using the Kruskal–Wallis test with Dunn’s multiple comparison test. * *p* < 0.05 and ** *p* < 0.01.

**Figure 4 biomedicines-09-00806-f004:**
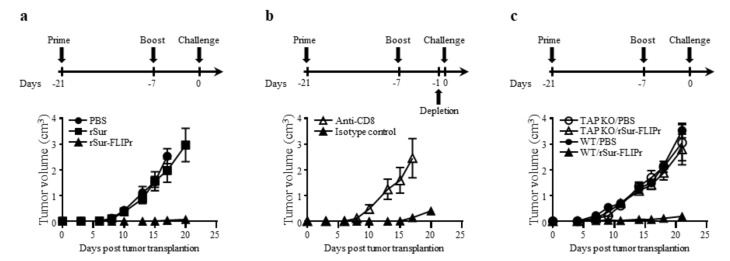
Immunization with rSur-FLIPr induces antitumor immunities through CD8- and TAP-dependent pathways. (**a**) Groups of C57BL/6 mice (*n* = 5) were immunized with rSur or rSur-FLIPr twice at a two-week interval (30 µg per dose). Mice injected with PBS served as the controls. (**b**) C57BL/6 mice were immunized with rSur-FLIPr twice at a two-week interval (30 µg per dose). One day before tumor implantation, the mice were randomly divided into two groups (*n* = 10) and then were intraperitoneally injected with anti-CD4, anti-CD8, or isotype control antibodies. (**c**) C57BL/6- or TAP-deficient mice (*n* = 5/group) were immunized with PBS or rSur-FLIPr (30 µg per dose) twice at a two-week interval. Seven days after the second immunization, the animals were subcutaneously inoculated with EG7 cells (5 × 10^4^/mouse). Tumor growth was monitored after challenge. The data are expressed as the means ± SEM.

**Figure 5 biomedicines-09-00806-f005:**
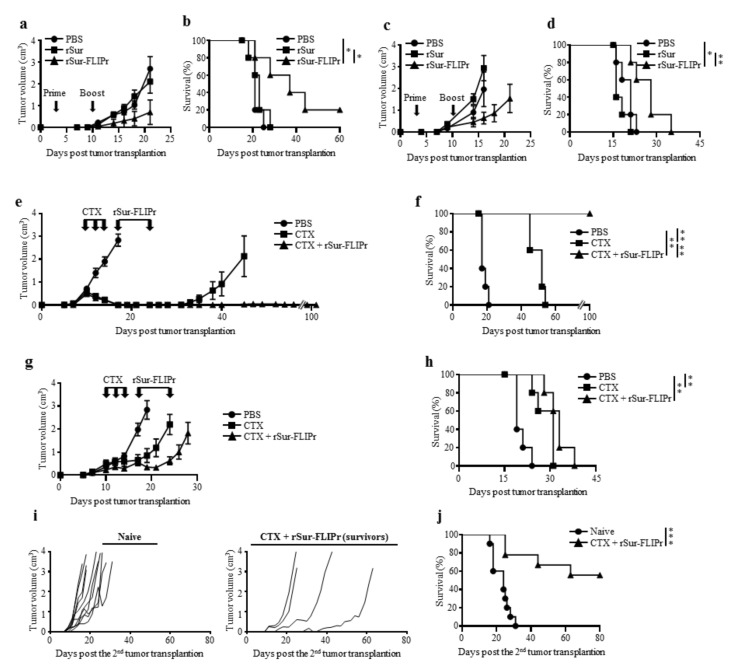
A combination of chemotherapy with rSur-FLIPr enhances the antitumor effects. Groups of C57BL/6 mice (*n* = 5) were subcutaneously inoculated with 5 × 10^4^ EG7 (**a**,**b**,**e**,**f**) or 1 × 10^5^ B16F10 (**c**,**d**,**g**,**h**) cells on day 0. The animals were treated with PBS, rSur, or rSur-FLIPr on day 3 and 10 (**a**,**b**,**c**,**d**). Tumor-bearing mice were treated with cyclophosphamide on day 10, 12, and 14 (3 mg per dose) plus rSur-FLIPr (30 µg per dose) or PBS on day 17 and 24 (**e**,**g**,**f**,**h**). Tumor growth (**a**,**c**,**e**,**g**) and survival rate (**b**,**d**,**f**,**h**) were monitored. The results are one of two representative experiments. After treatment with cyclophosphamide plus rSur-FLIPr, tumor-free mice (*n* = 9) that survived for 100 days were rechallenged with EG7 cells. Tumor growth (**i**) and survival rate (**j**) were monitored. The statistical significance was determined using the log-rank (MauteeCox) test. ** p* < 0.05, *** p* < 0.01, and **** p* < 0.001.

**Figure 6 biomedicines-09-00806-f006:**
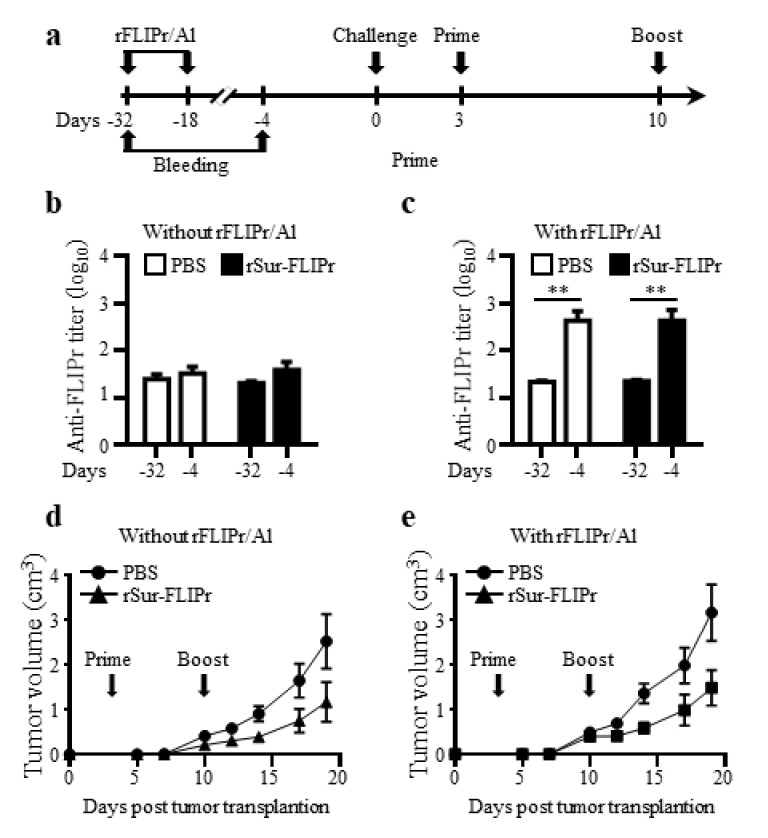
Preexisting anti-FLIPr antibodies do not diminish the antitumor capacity induced by rSur-FLIPr. The experimental flow chart is shown (**a**). Groups of C57BL/6 mice were immunized twice at a two-week interval by subcutaneous injection without/with rFLIPr plus aluminum on day -32 and -18. The mice in each group were randomly divided into 2 subgroups and treated with PBS or rSur-FLIPr at day 3 and 10 after EG7 inoculation. The sera were collected on day -32 (baseline) and -4 (after the second immunization). The anti-FLIPr antibody titers were determined by ELISA (**b**,**c**). The tumor volume was calculated as length × width × width/2 (mm^3^) (**d**,**e**). The data are expressed as the means ± SEM. The statistical significance was determined using the Mann–Whitney test. ** *p* < 0.01. The results are one of two representative experiments.

## Data Availability

Not applicable.

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
