# Peer review of "A Novel Recombinant Fcγ Receptor-Targeted Survivin Combines with Chemotherapy for Efficient Cancer Treatment"

_biomedicines, 2021, doi:10.3390/biomedicines9070806_

Round 1
Reviewer 1 Report
The manuscript submitted aims to demonstrate that FLIPr can be used as an antigen delivery vector and that rSur-FLIPr fusion protein is a possible therapeutic cancer vaccine.
The authors provide a clear introduction to the subject, stating the potential of FLIPr as a vector for antigen delivery, the role of survivin as a tumor-associated antigen and the relevance of using rSur-FLIPr fusion protein as a cancer vaccine.
The title chosen is consistent and appropriate considering the content.
The abstract provides an accessible summary, and the keywords reflect the content.
Overall, the manuscript is well structured. Each section has a conclusion that guides the reader through the steps taken by the authors to prove their hypothesis. It is clear in the introduction that the authors have previously published results (reference 11) that were taken in consideration and further developed in this manuscript.
The methods used and models chosen by the authors are appropriate and thoroughly described.
The figures effectively illustrate the results presented in the manuscript.
In the results section 3.4, the paragraph between line 324-333 is repeated in line 334-343.
In the discussion section line 409-410 the authors could clarify the statement “rSur-FLIPr is superior to rSur”.
The references are correct and fitting to the statements presented in the manuscript.
Minor correction in writing:
Line 176. “Intraperitoneal (i.p.)” instead of “i.p” (used for the first time in the text); from then on i.p. can be used (ex. line 188)
Line 275. “in all three groups of mice” instead of “in the all three groups of mice”
Reviewer 2 Report
It is interesting paper with potential clinical impact.
Reviewer 3 Report
It is a well written paper regarding the combination of chemotherapy with rSur-FLIPr for treatment to tumor bearing mice.
No comments.
